# Gravitational Waves from a Core-Collapse Supernova: Perspectives with Detectors in the Late 2020s and Early 2030s

**Marek Szczepańczyk** [1,*] and **Michele Zanolin** [2]

1 Department of Physics, University of Florida, Gainesville, FL 32611, USA
2 Physics and Astronomy Department, Embry-Riddle University, 3700 Willow Creek Road, Prescott, AZ 86301, USA; michele.zanolin@ligo.org
* Correspondence: marek.szczepanczyk@ligo.org

**Abstract:** We studied the detectability and reconstruction of gravitational waves from core-collapse supernova multidimensional models using simulated data from detectors predicted to operate in the late 2020s and early 2030s. We found that the detection range will improve by a factor of around two with respect to the second-generation gravitational-wave detectors, and the sky localization will significantly improve. We analyzed the reconstruction accuracy for the lower frequency and higher frequency portion of supernova signals with a 250 Hz cutoff. Since the waveform's peak frequencies are usually at high frequencies, the gravitational-wave signals in this frequency band were reconstructed more accurately.

**Keywords:** gravitational waves; core-collapse supernova; future detectors





## 1. Introduction

The observation of gravitational waves (GWs) from the binary black hole merger in 2015 [1] marks the beginning of GW astrophysics. In 2017, the binary neutron star was discovered in both GW and electromagnetic spectra [2], while in 2020, the GWs from two neutron star–black hole coalescences were observed [3]. To date, the Advanced LIGO [4] and Advanced Virgo [5] detectors have registered around 90 GW candidates with probability of astrophysical origin greater than 50% [6–8]. The newly born GW astrophysics has already challenged our understanding of the Universe, and we expect this trend will continue. With increased detectors sensitivity, the number of detections is anticipated to grow rapidly. Many GW sources are awaiting discovery, and core-collapse supernovae (CCSNe) are one of them. The next nearby CCSN will be one of the most interesting astronomical events of the century, and GWs will allow us to probe these phenomena's inner engines directly.

The prospects of exploring the Universe with GWs motivate us to improve and expand the current network of gravitational wave detectors. The advanced, second-generation (2G) detectors will reach their designed sensitivities in a few years, and the plans beyond 2G detectors are progressing. Recently the GW network expanded with KA GRA [9]. LIGO detectors could be substantially upgraded to the "LIGO Voyager" [10]. LIGO India [11] is currently under construction, and a neutron star extreme matter observatory (NEMO) detector in Australia is planned [12]. These advances will happen at the end of the 2020s and the beginning of the 2030s [13]. We dub them 2.5G detectors, as they are predicted to operate before the planned third-generation (3G) detectors, such as the Einstein Telescope [14] and Cosmic Explorer [15].

The considerations of exploring the detectability and reconstruction of GWs from CCSN with 2G detectors can be found in [16,17]. This paper focuses on the benefits of an extended 2.5G detectors network. We estimate how well a CCSN source can be localized in the sky.

We quantify how accurately the GW signals, particularly the low- and high-frequency features, can be reconstructed. Section 2 describes the methodology. Our findings are described in Section 3 and the conclusions are presented in Section 4.

## 2. Methodology

### 2.1. GW Detectors

The 2.5G network will consist of upgraded existing interferometers and the new detectors are planned for the late 2020s and early 2030s. It is difficult to make accurate predictions in a decade timescale, but we assume six detectors to be operating. Figure 1 shows the detectors' sensitivities considered in this paper. LIGO detectors in Hanford (H) and Livingston (L) will be upgraded to a Voyager sensitivity [10,18]. Virgo (V) will operate at an Advanced Virgo plus [9,19,20] sensitivity, and KA GRA (K) will reach its designed sensitivity [20]. LIGO India (A) is planned to operate at Advanced LIGO plus' sensitivity [20]. While the construction just began in Aundha in India, for simplicity, we assumed the arms to go along geographical directions of longitude and latitude. The NEMO (N) detector is an interferometer primarily targeting the postmerger GW signal from the binary neutron stars. It is optimized to be most sensitive in higher frequencies (around 2 kHz) than the other detectors [12]. This detector has been proposed, and for this study, we assumed its location and arm orientations to be that of AIGO [21]. We note that the location of the NEMO detector could be optimized for maximizing the results but this is beyond the scope of this work.

Given the different sensitivities of each detector, we considered a limited set of network configurations to study. The Voyager detectors are expected to be most sensitive; the networks containing the L and H detectors were considered: LHVKAN, LHVKA, LHVK, LHVK, LHV, LHA, LHK, LHN, LH. We also considered a network that did not contain the Voyager detector, VKA.

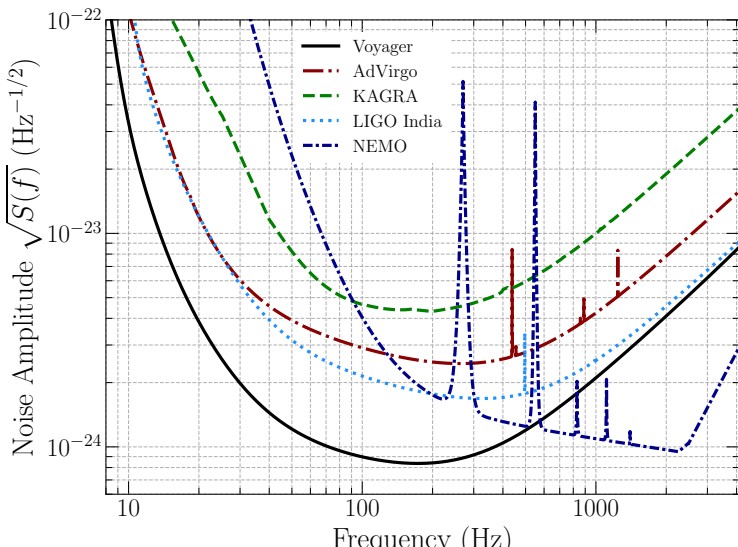

**Figure 1.** Sensitivities of the detectors projected to operate in the late 2020s and early 2030s. NEMO detector is designed to measure the high-frequency GW signal from binary neutron star postmerger.

### 2.2. Coherent WaveBurst

Coherent WaveBurst is an excess-power search algorithm for detecting and reconstructing GWs [22]. It does not assume any signal model and uses minimal assumptions on the signal morphologies. The analysis of GW strain data is performed in a wavelet domain [23]. Wavelets with amplitudes above the fluctuations of the detector noise are selected, grouped into clusters, and coherent events are identified. The cWB ranks events with an $\eta_c$ statistics based on the coherent network energy $E_c$ obtained by cross-correlating detectors data [22]. The $\eta_c$ is approximately the coherent network signal-to-noise ratio.

While it is not possible to predict a realistic detector noise reliably for the future detectors, we used a Gaussian noise recolored to the detector sensitivities depicted in Figure 1. In our analysis, we accepted events with a signal-to-noise ratio of 12 and those with a correlation coefficient $c_c$ of 0.7 ($c_c \approx 1$ means that an event is coherent across detectors and typically is a GW signal). These thresholds were used in [9]. The LIGO and Virgo detectors network are sensitive to detecting only one GW polarization component. Therefore, in the LIGO/Virgo/KA GRA searches (e.g., [24]) the cWB is typically tuned to reconstruct only one GW polarization component. The second polarization component can also be reconstructed for an extended network of GW detectors. Therefore, we loosened this constraint to allow a fraction of the second GW polarization component to be reconstructed as well.

### 2.3. CCSNe and GW Signals

CCSN is a violent death of a massive star (above $8\,M_\odot$). During its life, a star burns the fuel by means of nuclear fusion, creating an iron core in the center (see example reviews [25,26]). After exceeding the Chandrasekhar mass (around $1.4\,M_\odot$), the core collapses under the gravitational force forming a hot protoneutron star (PNS). Protons and electrons combine in such a dense environment, creating neutrons and neutrinos. In the so-called *neutrino-driven* mechanism (e.g., [27–32]), 99% of the CCSN energy is released with neutrinos, and they are believed to play a crucial role in the explosion (e.g., [26,33]). Typically, in this scenario, the GW emission comes from the oscillations of the PNS where the restoring force can be gravity, surface, or pressure (g-, f-, p-modes). During the explosion, the shock may also oscillate leading to the standing-accretion shock instability (SASI) [34].

In a case when a progenitor star rotates rapidly, a *magnetorotationally driven* (MHD-driven) mechanism, an explosion might be caused by energetic jets formed by a strong magnetic field (e.g., [35,36]). Alternatively, a failed CCSN may lead to a *black hole formation* (e.g., [37–39]).

The efforts to understand the CCSN engine are more than 50 years old [40]. Still, the explosion mechanism is not yet fully understood (see [25] for a review). Direct observation of GWs from a CCSN event will give us direct insight into the dynamics of this phenomenon. The theoretical and numerical CCSN simulations advanced significantly in recent years, and more accurate GW signal predictions are available. Our analysis used waveforms from multidimensional simulations, five for neutrino-driven explosions, one for MHD-driven explosions, and one for black hole formation.

Andresen et al., 2019 [30] studied the impact of moderate progenitor rotation on the GW signals. GW emission in the pre-explosion phase strongly depended on whether the SASI dominated the postshock flow with neutrino transport and g-mode frequency components in their signals. We used the m15nr model with a strong SASI below 250 Hz.

Kuroda et al., 2016 [41] studied the impact of the equation-of-state (EOS) on the GW signatures using a $15\,M_\odot$ progenitor star. We used the SFHx model with strong SASI at around 100 Hz, and the g-mode had a strong emission around 700 Hz.

Mezzacappa et al., 2020 [27] studied the origins of the GW emission for a $15\,M_\odot$ progenitor star (C15-3D model). The SASI/convection GW emission was present below 200 Hz, while the PNS oscillations occurred primarily above 600 Hz.

O'Connor and Couch, 2018 [31] analyzed the impact of the progenitor asphericities, grid resolution and symmetry, dimensionality, and neutrino physics for a $20\,M_\odot$ progenitor star. We explored the flagship mesa20 model. The resulting GW signals had strong SASI activity, but the g-mode dominated them.

Pan et al., 2020 [39] explored an impact of progenitor star rotation on an explosion and BH formation. We used the SR model for slowly rotating progenitor stars. The star exploded, and the fallback accretion led to a BH formation. The emitted GW energy was large, around $9 \times 10^{-7}\,M_\odot c^2$.

Radice et al., 2019 [29] studied GW emission depending on the progenitor star mass. The PNS oscillations usually dominated the GW signals. We studied the s13 model,

an explosion of a 13 $M_\odot$ progenitor star mass. Besides the PNS oscillations, a strong prompt convection below 100 Hz was present.

Richers et al., 2017 [35] analyzed core-bounce GWs for 18 EOSs from the axisymmetric rapidly rotating stars. We used the A467w0.50_SFHx model. The waveform was around 10 ms long and broadband.

## 3. Results

### 3.1. Detection Range

In this analysis, we studied the detectability of GWs as a function of a source distance. We took waveforms from different source angle orientations, rescaled their amplitudes to a given source distance and placed them randomly in the sky. The waveforms were added (injected) to the detector's data stream and reconstructed with cWB. The procedure was repeated for a range of the source distances and allowed us to create a detection efficiency curve as a function of distance. A distance at 50% detection efficiency is referred to as a *detection range*.

The left panel of Figure 2 shows the detection efficiency curves for all considered morphologies and the LHVKA network. The detectability of the most energetic waveforms, SR and A467w0.50_SFHx, exceeds 100 kpc. Less energetic waveforms are detectable at shorter distances. The right panel of Figure 2 considers the s13 model for nine network configurations. Because the noise floor for these two interferometers is the lowest among the 2.5G detectors, the detection distance for the networks with L and H detectors is approximately the same. The least sensitive network is VKA, while the most sensitive is the LHVKA network. The efficiency of the LHVKAN network decreases significantly at low distances, as the N detector is sensitive in a different frequency band than all other detectors. Because the s13 waveform is broadband, the detectors reconstruct different parts of the GW signal. This disbalance can break the coherence and often leads to a rejection of an event by cWB. In such a case, the cWB algorithm needs optimizing, but it is beyond the scope of this paper.

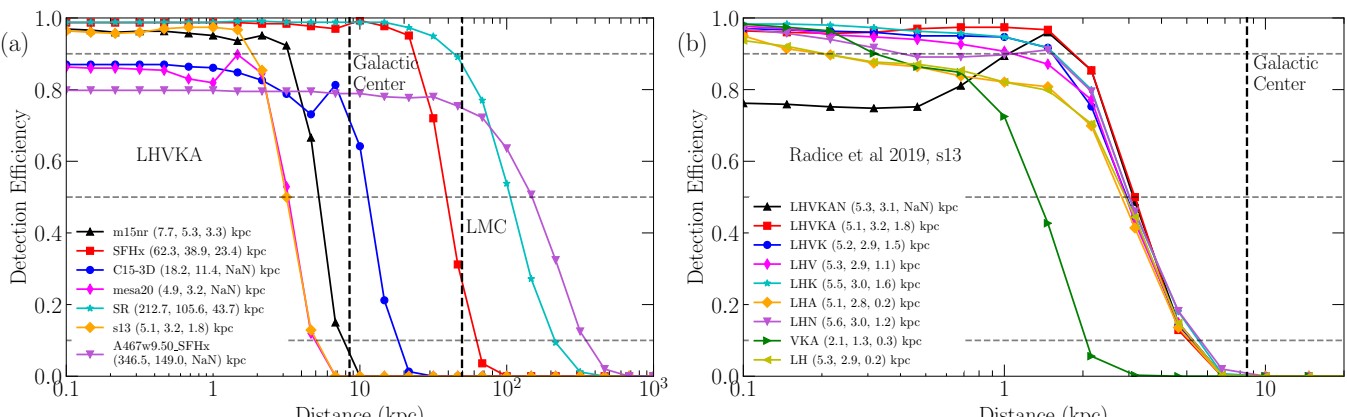

**Figure 2.** Detection efficiency curves as a function of distance. Numbers in brackets are distances at 10%, 50% and 90% detection efficiency. Plot (**a**) shows how far a CCSN source can be detectable for different models of GW emission [27,29–31,35,39,41] and LHVKA network, and (**b**) different detector networks for the s13 model [29].

Table 1 summarizes the results for all considered waveforms and network configurations. The short A467w0.50_SFHx waveforms are well detectable for all networks containing L and H. Similarly to s13, including NEMO leads to decreased efficiencies at shorter source distances. The least sensitive network is VKA. Compared with [16], the predicted detection ranges for the fifth observing run (O5) for m15nr, SFHx, C15-3D, mesa20, and s13 are 3.3, 21.6, 8.2, 2.0, and 1.8 kpc, respectively. The 2.5G detectors will improve these numbers roughly by a factor of two.

**Table 1.** Distances at 10%/50%/90% detection efficiency for all considered waveforms and detector networks.

| Network | Distance at 10%/50%/90% Efficiency (kpc) | | | | | | |
|---|---|---|---|---|---|---|---|
| | m15nr | SFHx | C15-3D | mesa20 | SR | s13 | A467w0.50_SFHx |
| LHVKAN | 8.6/5.5/- | 75.4/45.1/ 4.9 | 24.1/13.9/- | 7.5/0.4/- | 165.3/ 50.3/- | 5.3/3.1/- | 360.9/156.2/- |
| LHVKA | 7.7/5.3/3.3 | 62.3/38.9/23.4 | 18.2/11.4/- | 4.9/3.2/- | 212.7/105.6/43.7 | 5.1/3.2/1.8 | 346.5/149.0/- |
| LHVK | 7.8/5.0/2.3 | 60.2/36.9/19.3 | 17.6/10.7/1.2 | 4.6/2.8/1.3 | 210.6/109.1/32.7 | 5.2/2.9/1.5 | 354.2/122.8/- |
| LHV | 8.3/5.1/1.2 | 60.5/37.0/18.0 | 18.8/11.2/1.7 | 4.9/2.9/1.1 | 215.4/113.4/27.5 | 5.3/2.9/1.1 | 366.0/120.0/- |
| LHA | 8.4/5.2/2.4 | 60.8/37.9/21.1 | 19.3/11.9/1.8 | 5.4/3.3/1.5 | 218.2/110.9/27.8 | 5.5/3.0/1.6 | 360.6/130.6/- |
| LHK | 8.0/5.0/0.7 | 58.8/35.7/11.4 | 18.2/10.5/0.7 | 4.5/2.6/0.2 | 213.4/113.0/14.7 | 5.1/2.8/0.2 | 352.5/106.0/- |
| LHN | 9.1/5.7/1.9 | 70.8/42.6/20.9 | 26.6/12.9/1.3 | 8.3/4.8/0.5 | 183.6/ 48.2/ 8.5 | 5.6/3.0/1.2 | 378.3/130.2/- |
| VKA | 3.8/2.2/0.5 | 25.8/15.9/ 6.4 | 9.2/ 2.6/0.9 | 2.9/1.6/0.3 | 63.5/ 30.5/ 9.5 | 2.1/1.3/0.3 | 123.8/ 45.3/- |
| LH | 8.5/5.1/1.0 | 58.7/34.8/12.1 | 18.5/10.7/0.7 | 4.8/2.7/0.5 | 220.6/112.7/16.7 | 5.3/2.9/0.2 | 370.0/104.7/- |

## 3.2. Sky Localization

We studied how well the sky localization will improve with the 2.5G detectors. In this analysis, the waveforms from random source orientations were placed uniformly in volume and then reconstructed. The events were then reconstructed together with the estimated search area. The search area on the sky assigned a probability greater than or equal to the probability assigned to the injected location [42]. Figure 3 shows the search areas for a cumulative number of reconstructed events. The numbers in the brackets are areas at 0.1, 0.5, and 0.9 of the cumulative number of events.

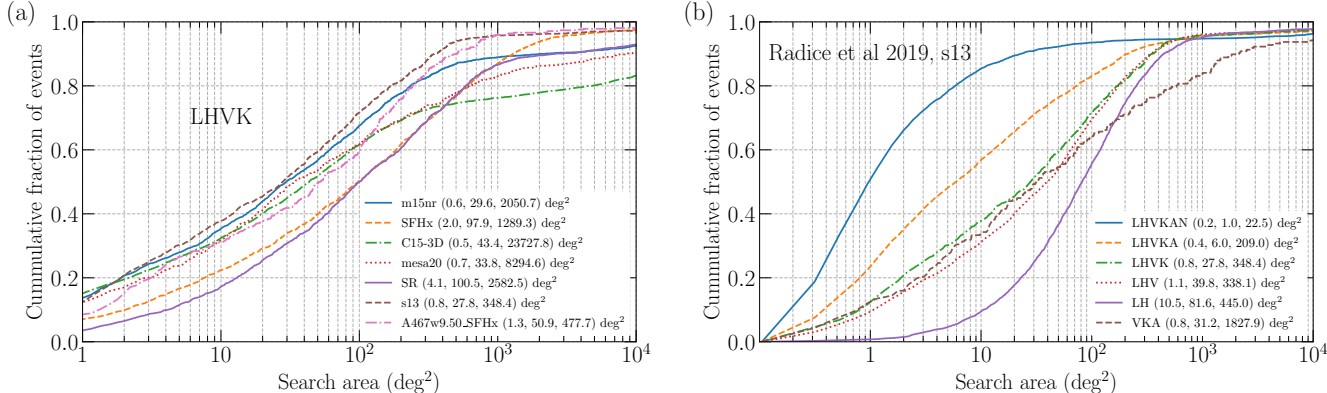

**Figure 3.** Search areas for (**a**) all GW signals [27,29–31,35,39,41] and (**b**) different detector networks for s13 model [29]. The search area is comparable between GW morphologies. Adding detectors to the network significantly reduces the search area. Numbers in brackets are 0.1, 0.5 and 0.9 of the cumulative number of events.

The left panel of Figure 3 shows the search area curves for all considered waveforms with the LHVK network. The search area curve depends on the waveforms, but it is comparable within the GW morphologies. Some neutrino-driven explosions with broadband GW signals, such as C15-3D, may lead to large search areas. The right panel of Figure 3 compares the search areas for the s13 waveform and different network configurations. The sky location of a GW source is calculated mainly based on the triangulation between the detectors. Clearly, adding more detectors decreases the search area. Comparable performance is seen for the LHV, LHVK, and VKA configurations. Table 2 summarizes the results for all waveforms and networks considered.

**Table 2.** Search areas for all waveforms and example detector networks.

| Waveform | Network | Area at 0.1/0.5/0.9 (deg$^2$) |
|---|---|---|
| m15nr | LHVK | 0.6/29.6/2050.7 |
| SFHx | LHVK | 2.0/97.9/1289.3 |
| C15-3D | LHVK | 0.5/43.4/23727.8 |
| mesa20 | LHVK | 0.7/33.8/8294.6 |
| SR | LHVK | 4.1/100.5/2582.5 |
| s13 | LHVKAN | 0.2/1.0/22.5 |
| s13 | LHVKA | 0.4/6.0/209.0 |
| s13 | LHVK | 0.8/27.8/348.4 |
| s13 | LHV | 1.1/39.8/338.1 |
| s13 | LHV | 10.5/81.6/445.0 |
| s13 | VKA | 0.8/31.2/1827.9 |
| A467w0.50_SFHx | LHVK | 1.3/50.9/477.7 |

*3.3. Reconstruction Accuracy*

The predicted GW signatures can be divided into low-frequency (LF), such as SASI/convection, and high-frequency (HF) signatures, such as the PNS oscillation. The division can be placed around 250 Hz. We quantified the accuracy of the cWB reconstruction for the full reconstructed waveform, HF, and LF waveform components. We used the waveform overlap, or a match [43] that was defined as a product between a detected $\mathbf{w} = \{w_k(t)\}$ and an injected whitened waveform $\mathbf{h} = \{h_k(t)\}$:

$$O(\mathbf{w}, \mathbf{h}) = \frac{(\mathbf{w}|\mathbf{h})}{\sqrt{(\mathbf{w}|\mathbf{w})}\sqrt{(\mathbf{h}|\mathbf{h})}} \,, \tag{1}$$

where $(.|.)$ denotes a scalar product in the time domain:

$$(\mathbf{w}|\mathbf{h}) = \sum_k \int_{t_1}^{t_2} w_k(t)h_k(t)dt. \tag{2}$$

The index $k$ is a detector number, and $[t_1, t_2]$ is the time range of the reconstructed event. The waveform overlap ranged from $-1$ (sign mismatch) to 1 (perfect reconstruction). The HF and LF waveform components were obtained using a high-pass and low-pass Butterworth filter with a 250 Hz cutoff frequency. Similarly to Section 3.2, we used uniform-in-volume injections.

Figure 4 shows a waveform overlap for the m15nr waveform that shows strong SASI and g-mode divided around 250 Hz. The reconstruction accuracy grows with the signal SNR. However, in this example, many waveforms have a poor reconstruction (16% with a waveform overlap below 0.2). The numbers in the brackets show an average waveform overlap at an SNR of 20 and 40, respectively. Only waveforms with a waveform overlap larger than 0.2 are selected. The waveform overlaps for the HF and LF waveform components are typically smaller than those for the full waveform. In this example, the HF component is reconstructed more accurately than the LF part of the signal. Because the waveform peak frequency is around 820 Hz, the HF part of the waveform is reconstructed first. The LF SASI is weaker and relatively long, and the accuracy of its reconstruction is worse.

Table 3 summarizes the results for all considered waveforms and networks. Typically, around 20% of the waveforms are misreconstructed (waveform overlap smaller than 0.2) for the neutrino-driven explosions and LHVK network. In around 56% of the BH formation, the SR waveforms are misreconstructed because the signal is almost 1 s long, and a small portion of the GW signal is reconstructed after the dominant prompt-convection. For the s13 waveform, the least misreconstructions are for the LH and LHV networks, as the cWB is optimized. As in the example of m15nr waveform, typically, the waveform is reconstructed around the peak frequency at low SNR, and reconstruction accuracy in that frequency range is better. Given

the peak frequencies of the considered waveforms is above 250 Hz, the waveform overlaps are typically higher in the HF part of the waveform. The LF SASI/convection are weaker parts of the waveforms, resulting in less accurate reconstruction. In the case of the SFHx, the spectrum has distinguished peak frequencies around 100 Hz and 700 Hz, resulting in an accurate reconstruction at low SNR for the HF and LF components. The accuracy decreases with higher SNR values as a more stochastic signal contributes.

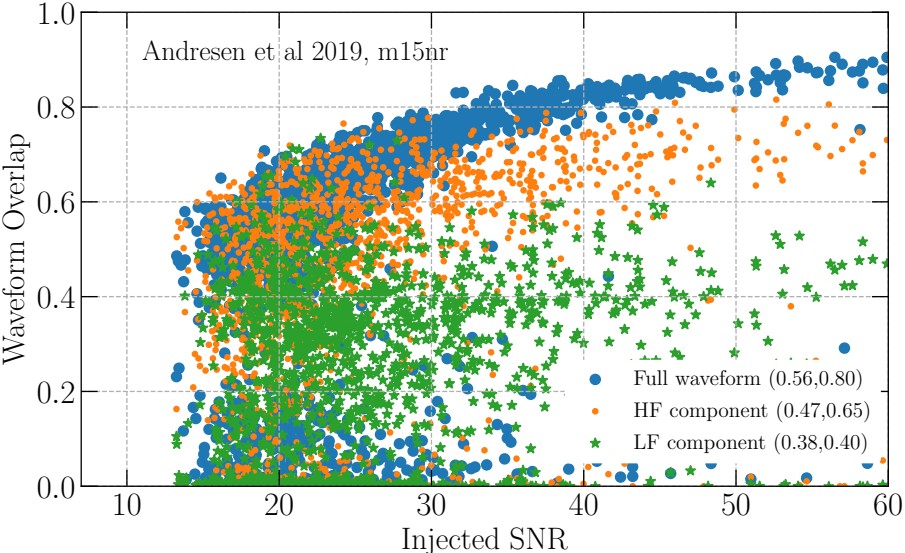

**Figure 4.** Waveform overlap as a function of injected SNR for m15nr model [30]. The accuracy of the full waveform increases with the SNR. However, in this example, each waveform component is reconstructed less accurately. The numbers in brackets are waveform overlaps at SNR 20 and 40, respectively, (for $O > 0.2$).

**Table 3.** Waveform overlaps for all waveforms and example detector networks. The reconstruction accuracy depends on the waveforms morphology and typically the waveform component is larger for the part of the signal with the peak frequency.

| Waveform | Network | Mis-Recon. $O < 0.2$ | O for Full/HF/LF Waveform ($O > 0.2$) SNR = 20 | SNR = 40 |
|---|---|---|---|---|
| m15nr | LHVK | 16% | 0.56/0.47/0.38 | 0.80/0.65/0.40 |
| SFHx | LHVK | 29% | 0.61/0.73/0.74 | 0.87/0.53/0.68 |
| C15-3D | LHVK | 18% | 0.36/0.40/0.22 | 0.64/0.57/0.23 |
| mesa20 | LHVK | 20% | 0.43/0.48/0.51 | 0.62/0.57/0.38 |
| SR | LHVK | 56% | 0.57/0.27/0.31 | 0.77/0.45/0.33 |
| s13 | LHVKAN | 24% | 0.60/0.59/0.39 | 0.73/0.72/0.37 |
| s13 | LHVKA | 23% | 0.66/0.67/0.39 | 0.81/0.82/0.40 |
| s13 | LHVK | 23% | 0.69/0.69/0.39 | 0.83/0.76/0.43 |
| s13 | LHV | 10% | 0.70/0.70/0.38 | 0.82/0.83/0.43 |
| s13 | LH | 7% | 0.73/0.73/0.39 | 0.86/0.81/0.44 |
| s13 | VKA | 28% | 0.59/0.61/0.34 | 0.65/0.40/ - |
| A467w0.50_SFHx | LHVK | 16% | 0.81/0.61/0.60 | 0.81/0.72/0.69 |

## 4. Conclusions

The era of GW astrophysics has begun, and our understanding of the dynamical Universe is advancing. CCSNe are one of the most interesting cosmic phenomena, and detecting GWs will shed light on understanding the nature of these explosions. The 2.5G detector network will extend around the detection range of around a factor of two, and this improvement will come primarily from the Voyager sensitivity. More detectors in the network will allow significantly improved sky localization of the source than with

the 2G detectors. This improvement is important when the galactic dust obscures an optical counterpart of a nearby event. We studied the reconstruction accuracy of the waveforms and their HF and LF components. Typically, the reconstruction was more accurate around the peak frequency of the signal, and the LF waveform component was reconstructed less accurately. The 2.5G detector network will better reconstruct both GW polarization states with respect to the 2G detector network. It will enable a better probing of the supernova's inner engine.

**Author Contributions:** Conceptualization, M.S.; formal analysis, M.S.; investigation, M.S. and M.Z.; writing—original draft preparation, M.S. and M.Z.; writing—review and editing, M.S. and M.Z. All authors have read and agreed to the published version of the manuscript.

**Funding:** The work by M.S. was supported by NSF grants no. PHY 1806165 and no. PHY 2110060. M.Z. was supported by NSF grant no. PHY 1806885.

**Institutional Review Board Statement:** Not applicable.

**Informed Consent Statement:** Not applicable.

**Data Availability Statement:** Not applicable.

**Acknowledgments:** This research has made use of data, software, and/or web tools obtained from the Gravitational Wave Open Science Center, a service of LIGO Laboratory, the LIGO Scientific Collaboration, and the Virgo Collaboration. This material is based upon work supported by NSF's LIGO Laboratory which is a major facility fully funded by the National Science Foundation. We thank David Radice for the insightful comments. This paper has LIGO document number P2200072.

**Conflicts of Interest:** The authors declare no conflict of interest.

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
