# Peer review of "Gravitational Waves from a Core-Collapse Supernova: Perspectives with Detectors in the Late 2020s and Early 2030s"

_galaxies, doi:10.3390/galaxies10030070_

Round 1

Reviewer 1 Report

The manuscript investigates the prospects of detecting gravitational waves from core-collapse supernovae (CCSNe) with the 2.5 generation detectors. Using the Coherent WaveBurst (CWB) algorithm, the authors study how distance and search area affect the detectability of different detector networks for various CCSN models. Then the accuracy of the algorithm is evaluated by calculating the waveform overlap between the reconstructed waveforms and the injected waveforms. I find the manuscript illustrative and well-written. I'm only puzzled by Eq. (2). As far as I understand, the inner product in the frequency domain involves the noise spectral density, I don't see how the dependence on the noise disappears when calculating the inner product in the time domain. It seems to me that using Eq. (2) to calculate the inner product is only valid for white noise, which is not the case for GW detectors.

I would recommend the manuscript for publication after the above concern is addressed. Also, there are two minor suggestions.  

1. I understand the abbreviation "PNS" is for proto-neutron star, not proton-neutron star. (line 82)

2. In figure 2, while it's said in the caption the numbers in the brackets are distances at 50% detection efficiency, those numbers in the plots are for 10%, 50%, and 90% detection efficiencies. The caption should be consistent with the plot legends.

Author Response

We thank the reviewer for reading carefully the manuscript and providing useful comments.

0. Comment regarding the Eq. (2):
> The reviewer is right and indeed the injected waveform involves the noise spectra, that are whitened. We added the word "whitened".

1. I understand the abbreviation "PNS" is for proto-neutron star, not proton-neutron star. (line 82)

> Correction is made.

2. In figure 2, while it's said in the caption the numbers in the brackets are distances at 50% detection efficiency, those numbers in the plots are for 10%, 50%, and 90% detection efficiencies. The caption should be consistent with the plot legends.

> Correction is made.

Reviewer 2 Report

In this paper the authors describe the reconstruction of gravitational waves produced by the core collapse of supernovae and its consequent explosion.

Using simulated data from future gravitational wave detectors the authors report an improvement up to a factor of two with respect to the second second-generation gravitational-wave detectors.

The authors give a detailed analysis of the detectability of the signals in the upcoming detectors. The results are clearly presented and the paper is well written. I thus consider the paper is suitable for publication in its present form in Galaxies.

Author Response

We thank the reviewer for reading carefully the manuscript and a kind review.

Reviewer 3 Report

Thank you for the interesting and accessible account of CCSN detection prospects with GWs. My comments are as follows:

Line 26: There are no definite plans to upgrade LIGO to Voyager or any other configuration beyond A+. It could happen, but it has not been decided. The phrase “are planned to be” should be replaced with “could be” or something similar.

Fig. 2: Numbers in brackets are (I assume) 10,50,90% efficiency rather than just 50% efficiency, or else I misunderstood something.

Line 154: When the events are placed uniformly in volume, this is presumably only up to some maximum distance (large enough so that it encloses the sensitive volume of the network). Then in Fig. 3, what is meant when the cumulative fraction of events reaches 1.0? Does that correspond to all the injected events (and therefore depends on the choice of maximum distance of injection), or is there a threshold applied based on SNR?

Conclusion (or elsewhere): Can the authors make statements about what will (or won’t) be learned from observing a galactic CCSN with 2.5G detectors versus 2G or 3G detectors? I take it from Fig. 2a that if a CCSN occurs in the galaxy while 2.5G detectors are running, then the observation (or non-observation) of GWs from this event will distinguish some explosion mechanisms from others. Or said another way, if one wants to be sure to observe GWs from a galactic CCSN for all the possible explosion mechanisms considered in the literature, one needs a 3G detector. Is that right?

The rest are minor comments on wording:
Line 1: “observation of the GWs” -> “observation of GWs”
Line 21: “these phenomena’ inner energies” -> “the inner energies [or dynamics?] of these phenomena”
Lines 129-130: maybe “are added (injected) to the detector’s data stream”, or “are combined with the detector’s noise”
Line 201: “began” -> “has begun”

Author Response

We thank the reviewer for reading carefully the manuscript and providing useful comments.

Line 26: There are no definite plans to upgrade LIGO to Voyager or any other configuration beyond A+. It could happen, but it has not been decided. The phrase “are planned to be” should be replaced with “could be” or something similar.

> We agree and made the correction.

Fig. 2: Numbers in brackets are (I assume) 10,50,90% efficiency rather than just 50% efficiency, or else I misunderstood something.

> Correction is made.

Line 154: When the events are placed uniformly in volume, this is presumably only up to some maximum distance (large enough so that it encloses the sensitive volume of the network). Then in Fig. 3, what is meant when the cumulative fraction of events reaches 1.0? Does that correspond to all the injected events (and therefore depends on the choice of maximum distance of injection), or is there a threshold applied based on SNR?

> Cumulative fraction of events reaching 1.0 mean all events. The event with the largest reconstructed sky area will correspond to 1.0. All the considered events are after the selection cuts are applied.

Conclusion (or elsewhere): Can the authors make statements about what will (or won’t) be learned from observing a galactic CCSN with 2.5G detectors versus 2G or 3G detectors? I take it from Fig. 2a that if a CCSN occurs in the galaxy while 2.5G detectors are running, then the observation (or non-observation) of GWs from this event will distinguish some explosion mechanisms from others. Or said another way, if one wants to be sure to observe GWs from a galactic CCSN for all the possible explosion mechanisms considered in the literature, one needs a 3G detector. Is that right?

> We thank for this comment. We added sentences in the conclusion: "More detectors in the network will allow significantly improved sky localization of the source than the 2G detectors. This improvement is important when the Galactic dust obscures an optical counterpart of a nearby event." and "The 2.5G detector network will better reconstruct both GW polarization states with respect to the 2G detector network. It will enable probing better the supernova's inner engine."

Line 1: “observation of the GWs” -> “observation of GWs”

> Correction is made.

Line 21: “these phenomena’ inner energies” -> “the inner energies [or dynamics?] of these phenomena”

> The phrase "these phenomena' inner engines", includes the energies and the dynamics (frequency evolution, dominant frequency, time of emission and other). We hope the current phrase can stay.

Lines 129-130: maybe “are added (injected) to the detector’s data stream”, or “are combined with the detector’s noise”

> Correction is made with "data stream"

Line 201: “began” -> “has begun”

> Correction is made.